# Replacing Concentrate with Yeast- or EM-Fermented Cassava Peel (YFCP or EMFCP): Effects on the Feed Intake, Feed Digestibility, Rumen Fermentation, and Growth Performance of Goats

**DOI:** 10.3390/ani13040551

**Published:** 2023-02-04

**Authors:** Pongsatorn Gunun, Anusorn Cherdthong, Pichad Khejornsart, Metha Wanapat, Sineenart Polyorach, Walailuck Kaewwongsa, Nirawan Gunun

**Affiliations:** 1Department of Animal Science, Faculty of Natural Resources, Rajamangala University of Technology Isan, Sakon Nakhon Campus, Sakon Nakhon 47160, Thailand; 2Tropical Feed Resources Research and Development Center (TROFREC), Department of Animal Science, Faculty of Agriculture, Khon Kaen University, Khon Kaen 40002, Thailand; 3Department of Agriculture and Resource, Faculty of Natural Resources and Agro-Industry, Chalermphakiat Sakon Nakhon Campus, Kasetsart University, Sakon Nakhon 47000, Thailand; 4Department of Animal Production Technology and Fisheries, Faculty of Agricultural Technology, King Mongkut’s Institute of Technology Ladkrabang, Bangkok 10520, Thailand; 5Department of Animal Science, Faculty of Technology, Udon Thani Rajabhat University, Udon Thani 41000, Thailand

**Keywords:** agro-industrial by-products, cassava peel, effective microorganisms, replacement, yeast

## Abstract

**Simple Summary:**

The use of agro-industrial by-products as ruminant feed will lower the production costs and improve livestock profitability. Cassava peel is a major by-product of processing cassava roots for food, starch, and animal feed. The abundant fiber and low crude protein in cassava peel present a nutritional challenge. The goal of this study was to use a microorganism (*Saccharomyces cerevisiae* or effective microorganism, EM) to increase the nutritive value of cassava peel, use it as a replacement for concentrate, and examine the effects on the feed intake, feed digestibility, rumen fermentation, and growth performance of goats. The findings indicated that the feed intake, feed digestibility, rumen fermentation parameters, volatile fatty acid profile, overall average daily gain, and feed efficiency were not significantly different among treatments. Therefore, yeast- or EM-fermented cassava peel can replace up to 50% of the concentrate without affecting the feed intake, feed digestibility, rumen fermentation, or growth performance. This can lower the feed cost per gain by up to 32%.

**Abstract:**

The goal of this study was to improve the nutritional value of cassava peel by using yeast (*Saccharomyces cerevisiae*) or effective microorganisms (EM), then use it as a replacement for concentrate, and examine the effects on the feed intake, feed digestibility, rumen fermentation, and growth performance of goats. The experimental design was a randomized complete block design (RCBD), and the dietary treatments were (1) concentrate, (2) replacement of the concentrate by yeast-fermented cassava peel (YFCP) at 50% and (3) replacement of the concentrate by EM-fermented cassava peel (EMFCP) at 50%. Twelve goats were given concentrate based on their treatments at a rate of 1.5% of their body weight. Rice straw was used as roughage and fed freely. It was found that the crude protein (CP) content of the cassava peel from 2.1% to 13.7–13.8% after 14 days of fermentation. Moreover, there were no significant differences between the treatments in terms of the feed intake, feed digestibility, ruminal pH, blood urea nitrogen concentration, volatile fatty acid profile, overall average daily gain, and feed efficiency. However, the cost of the feed per gain decreased when the YFCP or EMFCP was used instead of the concentrate. Based on the results of this experiment, it was possible to conclude that yeast or EM might be utilized as microorganisms to increase the nutritional value of cassava peel. Moreover, YFCP or EMFCP can replace concentrate by up to 50% without an impact on the feed intake, feed digestibility, rumen fermentation characteristics, and the growth performance; whereas, it can reduce the feed cost per gain up to 32%.

## 1. Introduction

Feed is the most expensive component of ruminant production, accounting for approximately 65–70% of total costs. Many attempts have been made to reduce the cost of feeding to its minimum. As part of these efforts to make ruminant production more sustainable, expensive feedstuffs are being replaced with cheaper and more plentiful by-products. Crop residues and agro-industrial by-products include a wide range of materials, such as rice straw, maize stover, casava peel, and pulp. These feeds often contain large amounts of cellulose and hemicellulose, both types of carbohydrates [1,2]. Therefore, using agro-industrial by-products as feed for ruminants has a low nutritional value due to their low energy and nitrogen content, low intake [3], and low digestibility. As a result, animal production is low [4]. Animal nutritionists have tried to improve the nutritional value of by-products from the food and agriculture industries that are fed to ruminants. Many strategies have been used to improve agricultural by-product utilization, such as grinding, soaking, pelleting, or treating with alkaline, acidic, or oxidative agents, as well as fermenting with microorganisms such as yeast, fungi, and/or their enzymes [5,6].

Cassava peel is the most abundant and has the potential to be utilized as a ruminant basal feed. It has 3.0% CP, 22.9% neutral detergent fiber (NDF), 16.7% acid detergent fiber (ADF), and 12.6% lignin [7]. However, cassava peel contains very high levels of antinutrients in the form of hydrogen cyanide (HCN). In order to reduce the antinutrients, cassava must be processed using various strategies, including oven-drying, crushing and sun-drying, and fermenting [8]. Fermentation has a better nutritional value because the catabolic microorganisms break down complex components into simple easily digestible ones. Therefore, fermentation has been extensively employed to enhance the nutrients of a feedstuff, particularly the protein content [9].

In order to increase the nutritional value of agro-industrial by-products, the approach of increasing protein in the diet by utilizing microbes, including yeast (*Saccharomyces cerevisiae*) or a mixture of microbes (effective microorganisms, EM), has been studied [10,11]. Yeast fermentation of cassava peel enhanced its protein content from 2.4% to 14.1% [12]. Suranindyah and Astuti [13] reported that dried fermented cassava peel can be used to replace wheat bran at less than 30% of the concentrate in lactating goats. Moreover, Kalio [14] indicated that cassava peel plus cassava and sweet potato forage had the highest gas volumes, short-chain fatty acids, and organic matter digestibility. However, the use of yeast-fermented cassava peel (YFCP) or EM-fermented cassava peel (EMFCP) as a concentrate replacement has not yet been investigated. Thus, the purpose of this study was to demonstrate how replacing the concentrate with YFCP or EMFCP affected the feed utilization, rumen fermentation, and growth performance in goats.

## 2. Materials and Methods

### 2.1. Preparation of the Yeast- or EM-Fermented Cassava Peel (YFCP or EMFCP)

The treatment involved cassava peel fermentation with yeast or EM from KYUSEI Co., Ltd., Saraburi, Thailand. Three main kinds of aerobic and anaerobic microbes are present in the EM, which include lactobacillus bacteria, yeasts, and/or fungi, as well as photosynthetic bacteria [15]. 

YFCP or EMFCP was prepared with the medium and the solution as follows: (1) after weighing 20 g of yeast or EM and 40 g of molasses into a flask, 400 mL of distilled water was added, and the mixture was then incubated for 1 h at room temperature; (2) we prepared the medium by mixing 54 g of urea and 72 g of molasses in 500 mL of distilled water, and then adjusted the pH of the medium solution using H_2_SO_4_ to obtain a final pH of 3.5–5; (3) we mixed (1) and (2) in a 1:1 ratio, followed by a 60-hour oxygen flush; (4) after 60 h, we thoroughly combined the yeast or EM medium solution with cassava peel at a ratio of 100 g:10 mL; (5) then, we allowed the product to ferment for 14 days before using it as a concentrate replacement. The diet was stored in the fermentation bucket to be added to the concentrate or used as a direct replacement (modifying the method of Sommai et al. [16])

### 2.2. Animals, Diets, and Experimental Design

The experiment was conducted over 60 days at the farm of the Faculty of Technology, Udon Thani Rajabhat University, Udon Thani, Thailand. A 60-day feeding trial evaluating the dietary replacement of concentrate with YFCP or EMFCP at 50% was carried out using a randomized complete block design (RCBD). Twelve male crossbred (Thai native × Anglo–Nubian) goats with live weights of 13±4 kg were divided into four blocks based on their homogeneous body weight (BW). Three dietary treatments were provided at random to the animals in each block. The feed ingredients and chemical compositions of the concentrate, YFCP, and EMFCP are presented in Table 1 and Table 2.

The goats received the concentrate diets at 1.5% BW and rice straw free-choice feedings at 07:00 and 16:00. Each goat was housed in its own well-ventilated pen and had constant access to clean fresh water and mineral blocks.

### 2.3. Data Collection and Chemical Analysis

The average daily gain (ADG) was calculated by weighing goats at the beginning BW, 30 days, and at the final BW (60 days). Every morning, both the offered and the refused feed were recorded and chemically analyzed. The feed conversion ratio (FCR) was calculated according to the following formula: FCR = [dry matter intake (g)/BW change]. On days 56–60 of the trial, about 500 g of feces were collected using the total collection method to conduct a digestibility test. Daily, fresh feces samples from each goat were pooled and chilled at 4 °C. Feed, refuse, and feces samples were ground (1-mm screen using the Cyclotech Mill; Tecator, Hoganas, Scania, Sweden) after being dried at 60 °C. The amounts of ash, CP [17], NDF, and ADF [17,18] were measured.

On day 60, blood samples were collected from the jugular vein at 0 and 4 h after feeding in order to determine blood urea nitrogen (BUN) [19]. Rumen fluid was taken through a stomach tube connected to a vacuum pump (about 60 mL) from each animal (about 60 mL) from each animal. The ruminal pH was instantly determined using a portable pH meter. Four layers of cheesecloth were used to filter the rumen fluid samples before being divided for NH_3_-N measurement. The NH_3_-N concentration was measured with a Kjeltech Auto 1030 Analyzer, Tecator, Hoganiis, Sweden [20], and the volatile fatty acids (VFA) were assessed using high-performance liquid chromatography (HPLC; Model Water 600; UV detector, Millipore Corp., Milford, MA, USA) [21].

### 2.4. Statistical Analysis

All data were tested for normal distribution using the UNIVARIATE procedure in SAS software (Version 8, SAS Institute Inc.: Cary, NC, USA) and subjected to an analysis of variance using the GLM procedure [22]. The data were analyzed using the model Yi = µ + αi + βj + εij, where Yi is the dependent variable, µ is the overall mean, αi is the treatment effect (i = 1 to 3; concentrate, YFCP, or EMFCP), βj is the block effect (j = 1 to 4), and εij is the residual error. Tukey’s test was used to indicate the variations between the different treatments, and *p* < 0.05 was used to define the significant differences.

## 3. Results

### 3.1. Chemical Composition of Diets 

The feed ingredients and chemical composition are presented in Table 1 and Table 2. The CP content of the cassava peel, YFCP, and EMFCP were 2.1%, 13.7%, and 13.8%, respectively, while the NDF were 52.1%, 36.7%, and 35.1%. The concentrate contained 14% CP, 34.2% NDF, and 21.6% ADF. The chemical composition of the concentrate mixed with the YFCP and EMFCP at a ratio of 50:50 contained 13.9% CP, 35.8 and 35.9% NDF, and 22.8 and 23.0% ADF, respectively. Additionally, the rice straw contained 2.3% CP, 88.3% NDF, and 54.7% ADF.

### 3.2. Feed Intake and Nutrient Digestibility

The replacement of the concentrates with the YFCP or EMFCP had no effect on the roughage and total DM intake (*p* > 0.05) (Table 3). Likewise, the apparent digestibility was similar between the groups (*p* > 0.05). 

### 3.3. Rumen Fermentation and Blood Metabolites

There were no significant differences in the rumen pH among the treatments (*p* > 0.05). Furthermore, replacing the concentrate with YFCP or EMFCP did not change the NH_3_-N and BUN concentrations (*p* > 0.05). The concentrations of ruminal NH_3_-N and BUN were 17.2 to 21.7 mg/dL and 14.5 to 19.5 mg/dL, respectively (Table 4).

### 3.4. Volatile Fatty Acid (VFA) Profiles

The profiles of the volatile fatty acids (VFA) are presented in Table 5. The concentrations of total VFA (0 and 4 h post feeding), acetic acid, propionic acid, and butyric acid, as well as the acetic/propionic ratio, were unaffected by the replacement of the concentrate with YFCP or EMFCP (*p* > 0.05). However, the mean value of the total VFA was significantly lower (*p* < 0.05) when the goats were fed with the EMFCP.

### 3.5. Growth Performance

On days 30 and 60, the change in BW was similar among the treatments (*p* > 0.05). Similarly, replacing the concentrates with YFCP or EMFCP had no impact on the ADG (0 to 60 d) and feed efficiency (*p* > 0.05; Table 6). However, the YFCP-fed goats showed a greater ADG from days 31 to 60. Moreover, when YFCP or EMFCP were added to the goat feed, the ADG on days 0–30 and the cost of feed per kg of weight gain were significantly lower (*p* < 0.05).

## 4. Discussion

### 4.1. The Chemical Composition of the Diets 

The fermentation of cassava peel by yeast or EM can improve its nutritive value, especially by increasing its CP content, which could serve as an alternative source of protein for small ruminants. The increase in CP may be attributable to the microorganism’s ability to excrete enzymes into the substrate as a result of its metabolic activities, thereby promoting its growth. Thus, microorganism cells could be an additional source of microbial proteins [23]. Under this study, the CP content was higher than that reported by Suranindyah and Astuti [13]. Meanwhile, the *S. cerevisiae* fermentation of cassava pulp enhanced its protein content from 4.8% to 13.3% [24] or 23.3% [16]. This could be caused by many factors, including the source of carbohydrates and non-protein nitrogen (NPN) such as urea, their concentrations, the length of time incubated, and the type of microorganisms. Moreover, the NDF and ADF content of the YFCP decreased by 28% and 24%, while EMFCP decreased by 33% and 21%, respectively, when compared with the untreated cassava peel. The lower fiber content in the cassava peel fermented with yeast or EM could be attributed to the increased chemical hydrolysis of hemicelluloses by yeast or some bacteria, such as *Bacillus* sp., which are found in EM solutions and have the ability to extensively utilize the cellulose and lignin components [25]. 

### 4.2. Feed Intake and Nutrient Digestibility

In general, it is well known that feed consumption increases with increasing dietary CP content [26,27]. Wanapat et al. [28] confirmed the increase in the bacterial population, especially the rumen cellulolytic, amylolytic, and proteolytic bacteria, when SBM was replaced in the concentrate by yeast-fermented cassava chips, thereby increasing the feed intake and nutrient digestibility. The results of this study show that fermentation raised the CP content of the cassava peel from 2.1% to 13.7%–13.8% DM. In addition, when mixed with the concentrate in a 50:50 ratio of concentrate to YECP or EMFCP, the CP content (13.9% DM) was comparable to the CP content of the concentrate (14.0% DM). This could mean that the isonitrogenous between the dietary treatments did not alter the feed intake and digestibility. 

### 4.3. Rumen Fermentation and Blood Metabolites

Ruminal pH is the most important fermentation factor that directly influences the microbial ecology and ruminal fermentation [29]. In this study, the rumen pH ranged from 6.8 to 7.2, and the optimal range for microbial activity is between 6.5 and 7.0 [30]. 

According to Gunun et al. [31], NH_3_-N is the main nitrogen source for protein synthesis in the rumen. The ruminal NH_3_-N concentrations ranged from 17.2 to 21.7 mg/dL, which was close to the optimal range (15 to 30 mg/dL) [32,33]. In the present study, neither YFCP nor EMFCP affected the ruminal NH_3_-N concentration, indicating that the protein degradation by the rumen microbes was unaffected by the fermented cassava peel.

Protein status is frequently determined using the BUN concentration [34]. In general, a low BUN concentration indicates a limited protein intake or chronic hepatocellular disease. In contrast, an increase in BUN could be the result of renal failure and dehydration [35]. Our results show that replacing concentrate with YFCP or EMFCP had no effect on the BUN, which was between 14.5 and 19.5 mg/dL, within the normal range of 18.4 to 22.5 mg/dL in Thai-native × Anglo–Nubian goats [35].

### 4.4. Volatile Fatty Acid (VFA) Profiles

The molar proportion of the VFA, such as acetate, propionate, and butyrate, did not change when the YFCP or EMFCP replacement concentrate was added. The mean concentrations of the total VFA, acetic acid, propionic acid, and butyric acid were 101.5 to 110.4 mM/L, 65.3 to 67.6 mol/100 mol, 20.4 to 22.3 mol/100 mol, and 11.8 to 12.8 mol/100 mol, respectively. The findings showed that fermenting the cassava peel with yeast or EM may enhance quality and make it comparable to concentrate. This was similar to Norrapork et al. [36], who showed that fermenting cassava pulp with yeast may improve its nutritional value, especially when used with a 16% CP concentrate to obtain the highest levels of total VFA and propionic acid in cattle.

### 4.5. Growth Performance 

In the current study, the YFCP or EMFCP replacement concentrates had no effect on BW change or feed efficiency in goats (Table 6). In addition, although the ADG on days 0–30 was lower with the YFCP or EMFCP replacement concentrates, the ADG on days 31–60 was greater with the YFCP. Moreover, on days 0–60, the ADG was not significantly different when compared to the control group. The change in the ADG over time may be attributable to the animal’s adaptation to different diets. The ADG of the goats ranged from 43.3 to 60.0 g/day, 73.3 to 83.3 g/day, and 62.7 to 66.5 g/day on days 0–30, 31–60, and 0–60, respectively. Chobtang et al. [37] indicated that the final BW, weight gain, and ADG of Thai native × Anglo–Nubian crossbred goats increased when the CP levels in the total mixed ration (TMR) increased from 8% to 14%. Thus, it was demonstrated that replacing 50% of the concentrate with YFCP or EMFCP provided goats with a comparable nutritional growth to the concentrate feed. In contrast, replacing concentrate with YFCP or EMFCP has been demonstrated to reduce the goat feed costs per 1 kg increase in live weight by up to 32%. This showed that YFCP or EMFCP could be used as a 50% replacement for concentrate without affecting the overall ADG or feed efficiency. This would lower the cost of feed for Thai-native × Anglo–Nubian crossbred goats.

## 5. Conclusions

Replacement of concentrate in goat diets with YFCP or EMFCP had no adverse effect on the feed intake, feed digestibility, rumen characteristics, VFA profiles, or growth performance. On the other hand, it can reduce the cost of goat feed per kilogram of live weight gain. In order to develop the goat system and lower the cost of production, the YFCP or EMFCP can be used as a replacement for other protein sources or concentrate. More investigation is required to determine the effects of YFCP or EMFCP on goat carcass characteristics and meat quality.

## Figures and Tables

**Table 1 animals-13-00551-t001:** Ingredients and chemical compositions of the experimental diets.

Item	Concentrate	YFCP	EMFCP	Cassava Peel	RS
Ingredient, kg dry matter (DM)					
Cassava chip	47.0				
Rice bran	25.0				
Soybean meal	7.5				
Palm kernel meal	15.2				
Urea	1.8				
Molasses	1.5				
Minerals and vitamins	1.0				
Pure sulfur	0.5				
Salt	0.5				
Chemical composition					
Dry matter, %	-	36.3	36.8	29.6	91.6
Organic matter, %DM	-	93.9	92.9	83.2	93.1
Crude protein, %DM	-	13.7	13.8	2.1	2.3
Neutral detergent fiber, %DM	-	37.6	35.1	52.1	88.3
Acid detergent fiber, %DM	-	23.4	24.1	30.6	54.7

YFCP = yeast-fermented cassava peel; EMFCP = effective-microorganisms-fermented cassava peel; RS = rice straw.

**Table 2 animals-13-00551-t002:** Chemical compositions of the experimental diets.

Item	Control	Concentrate: YFCP 50:50	Concentrate: EMFCP 50:50
Chemical composition			
Dry matter, %	87.6	63.4	63.2
Organic matter, %DM	94.5	95.2	94.9
Crude protein, %DM	14.0	13.9	13.9
Neutral detergent fiber, %DM	34.2	35.8	35.9
Acid detergent fiber, %DM	21.6	22.8	23.0

YFCP = yeast-fermented cassava peel; EMFCP = effective-microorganisms-fermented cassava peel.

**Table 3 animals-13-00551-t003:** Feed intake and the apparent digestibility of the experimental diets in goats.

Items	Experimental Diets			SEM	*p*-Value
	Control	YFCP	EMFCP		
Roughage DM intake					
g/d	221.1	212.8	210.7	3.70	0.57
%BW	1.3	1.2	1.2	0.09	0.24
Concentrate DM intake					
g/d	198.2 ^a^	98.2 ^b^	96.7 ^b^	3.98	0.009
%BW	1.5 ^a^	0.75 ^b^	0.75 ^b^	0.01	0.002
Fermented cassava peel					
g/d	0.0 ^b^	97.8 ^a^	95.5 ^a^	6.37	0.008
%BW	0.0 ^b^	0.75 ^a^	0.75 ^a^	0.02	0.004
Total DM intake					
g/d	419.7	409.5	401.3	6.08	0.59
%BW	2.8	2.7	2.7	0.07	0.32
Apparent digestibility, %					
Dry matter	65.3	63.5	73.1	2.98	0.65
Organic matter	68.3	66.2	75.6	2.97	0.81
Crude protein	61.3	60.6	59.9	1.10	0.18
Neutral detergent fiber	57.5	56.8	56.9	2.24	0.73
Acid detergent fiber	55.2	54.9	55.1	2.33	0.92

^a,b^ Values on the same row with different superscripts differed (*p* < 0.05); YFCP = yeast-fermented cassava peel; EMFCP = effective-microorganisms-fermented cassava peel.

**Table 4 animals-13-00551-t004:** Rumen fermentation and the blood metabolites of the experimental diets in goats.

Items	Experimental Diets			SEM	*p*-Value
	Control	YFCP	EMFCP		
Ruminal pH					
0 h post feeding	7.1	7.1	7.2	0.04	0.46
4 h post feeding	6.9	6.9	6.8	0.02	0.53
mean	7.0	7.0	7.0	0.01	0.88
NH_3_-N concentration, mg/dL					
0 h post feeding	17.2	17.9	17.2	0.72	0.85
4 h post feeding	20.1	21.3	21.7	0.22	0.39
mean	18.7	19.3	19.6	0.54	0.71
Blood urea-N concentration, mg/dL					
0 h post feeding	14.5	15.8	17.0	1.07	0.73
4 h post feeding	17.3	18.3	19.5	1.30	0.99
mean	15.9	17.1	18.3	1.13	0.76

YFCP = yeast-fermented cassava peel; EMFCP = effective-microorganisms-fermented cassava peel.

**Table 5 animals-13-00551-t005:** The ruminal volatile fatty acid (VFA) profile of the experimental diets in goats.

Items	Experimental Diets			SEM	*p*-Value
	Control	YFCP	EMFCP		
Total VFA, mmol/L					
0 h post feeding	105.3	96.7	94.0	3.39	0.62
4 h post feeding	114.8	111.7	108.9	3.21	0.57
Mean	110.4 ^a^	104.2 ^ab^	101.5 ^b^	2.04	0.04
VFA, mol/100 mol					
Acetic acid					
0 h post feeding	69.1	69.5	70.8	1.72	0.63
4 h post feeding	61.3	64.9	64.5	1.55	0.44
Mean	65.3	67.2	67.6	1.51	0.70
Propionic acid					
0 h post feeding	19.2	18.8	18.3	1.62	0.62
4 h post feeding	25.3	22.2	22.6	1.79	0.39
Mean	22.3	20.5	20.4	1.31	0.47
Butyric acid					
0 h post feeding	11.7	11.6	10.9	0.36	0.71
4 h post feeding	12.9	13.9	12.8	0.62	0.51
Mean	12.3	12.8	11.8	0.51	0.62
Acetic/propionic acid ratio					
0 h post feeding	3.6	3.7	3.8	0.22	0.41
4 h post feeding	2.4	2.9	2.8	0.26	0.37
Mean	3.0	3.3	3.3	0.21	0.32

^a,b^ Values on the same row with different superscripts differed (*p* < 0.05); YFCP = yeast-fermented cassava peel; EMFCP = effective-microorganisms-fermented cassava peel.

**Table 6 animals-13-00551-t006:** Growth performance of the goats among the experimental diets.

Items	Experimental Diets			SEM	*p*-Value
	Control	YFCP	EMFCP		
Body weight, kg					
Initial	13.3	13.7	13.1	0.51	0.72
30 d	15.1	15.0	14.6	0.61	0.87
Final	17.3	17.5	16.8	0.42	0.54
ADG, g/d					
0 to 30-d	60.0 ^a^	43.3 ^c^	50.0 ^b^	1.65	0.03
31 to 60-d	73.3 ^b^	83.3 ^a^	73.3 ^b^	1.22	0.03
0 to 60-d	66.5	63.2	62.7	1.49	0.31
FCR					
0 to 30-d	6.7	6.9	7.0	0.66	0.45
31 to 60-d	5.9	6.1	6.0	0.23	0.69
0 to 60-d	6.3	6.5	6.5	0.45	0.34
Feed cost/1 kg of gain weight (USD/kg)	1.46 ^a^	1.02 ^b^	0.99 ^b^	0.09	0.001

^a,b,c^ Values on the same row with different superscripts differed ( *p* < 0.05). YFCP = yeast-fermented cassava peel; EMFCP = effective-microorganisms-fermented cassava peel.

## Data Availability

Not applicable.

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
