# Peer review of "Replacing Concentrate with Yeast- or EM-Fermented Cassava Peel (YFCP or EMFCP): Effects on the Feed Intake, Feed Digestibility, Rumen Fermentation, and Growth Performance of Goats"

_animals, 2023, doi:10.3390/ani13040551_

Round 1

Reviewer 1 Report

It should be considered that the use of these agro-industrial by-products must determine the processing to which they were submitted and consequently the response they have in the animals that consume them. also before assessing the response in carcass characteristics, and meat quality. hows on feed intake, weight gain and conversion, and feed efficiency.

Author Response

  1. It should be considered that the use of these agro-industrial by-products must determine the processing to which they were submitted and consequently the response they have in the animals that consume them. also before assessing the response in carcass characteristics, and meat quality. hows on feed intake, weight gain and conversion, and feed efficiency.
  • Cassava peel is an abundant agro-industrial byproduct that has the potential to be used as animal feed. In this study, the nutritional value of cassava peel can be improved by adding yeast (Saccharomyces cerevisiae) or effective microorganisms (EM) to increase CP and decrease fiber content. As revealed by the findings, yeast or EM-fermented cassava peel can replace up to 50 % of the concentrate without affecting feed intake, digestibility, rumen fermentation, or growth performance. This can reduce feed costs by up to 32% per kilogram of gain. However, the decision regarding its utilization is dependent on the region's availability of this residue and the cost of acquiring it.

Reviewer 2 Report

Dear Editor and Authors,

The use of agricultural byproducts in animal feed is an integral part of the circular economy, which is why such studies are of greater importance to me. A clear definition of the gap in the literature or the unanswered question that motivated this study is provided. A brief summary of the study's objective and approach is given in the last paragraph.  

I believe, however, that a sufficient number of animals is not available to produce a healthy outcome. Further, there are a few major shortcoming in other sections that need to be really fixed. I will be making general statements for the specified section since the pdf file does not contain line numbers.

Best regards,

Title, Simple summary and Abstract:

In spite of the title of the study encompassing more than one byproduct, the study aims to investigate the effects of fermented cassava peel that has been inoculated with a yeast or a mixture of inoculants (as stated "effective microorganisms" in the text). Therefore, it is necessary to change the title.

It would be helpful if you could indicate how many days the cassava peels have left to fermentation in Abstract.

Please sort the keywords alphabetically.

Introduction:

Please paraphrase this sentence "Several studies have reported on treatments using physics, chemistry, and biology".

Please use “antinutrients” instead of anti-nutrition in the second paragraph.

Materials and Methods:

Is it "sugar" or "molasses"? Which one did you use for your work? It is important to state clearly which one you used in Section 2.1.

As mentioned in the modified method, the fermented products were dried in the sun and the water ratio was reduced. Was drying done in this study? The statement needs to be clarified.

What is meant by "effective microorganisms"? Is the microorganism used homofermentative or heterofermentative, or a combination of both? What is the source of the product, what microorganisms are present, and what is the strain of microorganism?

It is necessary to specify the period during which the work was performed.

A sufficient number of animals is not available to produce a healthy outcome.

A VFA analysis requires specific information on the device, column, and features used in the chromatographic analysis.

Results:

In the ideal case, Table 5 should display the cost of feed in dollars at the exchange rate at the time the study was conducted.

Discussion:

Yeasts and some bacteria, such as Lentiloactobacillus buchneri, are capable of degrading cell walls. This needs to be referred to in Section 4.1.

It is not stated in the material method that the diets were prepared in an isonitrogenous form. There is also no way to mix 1:1 ratio in order to obtain an isonitrogenous diet compared to the control group (concentrate), as the concentrate feed and fermented cassava peel must have the same chemical composition in order to achieve this.

In section 4.3, it is recommended to omit the statement: " The results indicated that giving goats YFCP or EMFCP instead of concentrate had no effect on the rumen ecology or fermentation in goats". Just by looking at the pH of the rumen, we cannot conclude that it does not affect rumen ecology or fermentation.

Author Response

Title, Simple summary and Abstract:

  1. In spite of the title of the study encompassing more than one byproduct, the study aims to investigate the effects of fermented cassava peel that has been inoculated with a yeast or a mixture of inoculants (as stated "effective microorganisms" in the text). Therefore, it is necessary to change the title.
  • Already changed title to “Replacing Concentrate with Yeast or EM-Fermented Cassava Peel (YFCP or EMFCP) on Feed Intake, Digestibility, Rumen Fermentation, and Growth Performance in goats”, please see in the text.
  1. It would be helpful if you could indicate how many days the cassava peels have left to fermentation in Abstract.
  • Already added information, please see in line 38.
  1. Please sort the keywords alphabetically.
  • Already changed, please see in line 46.

Introduction:

  1. Please paraphrase this sentence "Several studies have reported on treatments using physics, chemistry, and biology".
  • Already improved, please see in line 58.
  1. Please use “antinutrients” instead of anti-nutrition in the second paragraph.
  • Already changed, please see in line 65.

Materials and Methods:

  1. Is it "sugar" or "molasses"? Which one did you use for your work? It is important to state clearly which one you used in Section 2.1.
  • Already checked and verified, please see in line 90.
  1. As mentioned in the modified method, the fermented products were dried in the sun and the water ratio was reduced. Was drying done in this study? The statement needs to be clarified.
  • In the present study, we use fermented products as wet form but, in fermentation process we modified the method of Sommai et al. (2022)
  1. What is meant by "effective microorganisms"? Is the microorganism used homofermentative or heterofermentative, or a combination of both? What is the source of the product, what microorganisms are present, and what is the strain of microorganism?
  • Already added information, please see in line 85.
  1. It is necessary to specify the period during which the work was performed.
  • Already added information, please see in line 101.
  1. A sufficient number of animals is not available to produce a healthy outcome.
  • Four animals per treatment (four replications) are often employed to minimize the number of animals required to detect statistical differences in a RCBD and CRD. Many ruminant experiments with four replications were published in a high-quality international journal. For example:
  1. Vivasane, S.; Preston, T.R. 2016: Effect of cassava (Manihot esculentaCrantz) and Erythrina (E. subumbrans) foliage on growth of goats fed basal diets of banana (Musa spp) leaves or Elephant grass (Pennisetum pupureum). Livest. Res. Rural. Dev. 2016, 28, Retrieved January 19, 2023, from http://www.lrrd.org/lrrd28/12/khao28215.html.

Abstract: The objective of this study was to measure the relative nutritive value of two protein-rich shrubs (Cassava and Erythrina subumbrans) fed as supplements to locally available forages (Elephant grass and banana leaves). Sixteen male local goats aged between 4 and 6 months and with a mean live weight of 10.2±2.3 kg were housed in individual cages according to a 2x2 factorial arrangement of 4 treatments: cassava or Erythrina as supplements; and banana leaves or Elephant grass as basal diet. There were 4 replications of each treatment.

  1. Silivong, P.; Preston, T.R. Growth performance of goats was improved when a basal diet of foliage of Bauhinia acuminatawas supplemented with water spinach and biochar. Livest. Res. Rural. Dev. 2015, 27, Retrieved January 19, 2023, from http://www.lrrd.org/lrrd27/3/sili27058.html.

Abstract: The aim of the study was to determine the effect of water spinach and biochar on enteric methane emissions, feed intake, digestibility, nitrogen retention and growth performance in local goats fed Bauhinia acuminata and molasses as basal diet. The experiment was arranged as a 2x2 factorial with 4 replications using sixteen goats in individual pens (initial live weight 12.9 kg and 6-7 months of age).

  1. Olajide, P.O.; Adeloye A.A. Values of microbial population in west African Dwarf goats fed on corncob-based concentrate diet with cobalt supplementation. Afr. J. Food Agric. Nutr. Dev. 2018, 1, 1-4.

Abstract: ABSTRACT: Inadequate feed supply and feeding is a major challenge in animal production. Corncob, an abundant by-product of maize production is seldom utilized for feeding purpose, though promising for ruminants. In a 90-day feeding trial conducted to study the response of West African Dwarf goats to corncob-based diet supplemented with varied levels of cobalt chloride. The study examined the effect of treatment on microbial population. Twelve West African Dwarf goats of similar body weights (8.42kg-8.51kg) and body conditions were allotted to three dietary treatment groups, of four replicates each in a randomized complete block design experiment.

  1. Pudasaini, R.; Dhital, B. Effect of Supplementing Rice Bran and Wheat Bran with Probiotics on Growth Performance of Khari Kids. Int. j. appl. sci. biotechnol2017, 5, 430–433.

Abstract: An experiment was conducted in IAAS, Livestock Farm, Rampur Chitwan, Nepal in order to assess the growth performance of kids feeding diets supplemented with probiotics. The experiment was laid out in Randomized Complete Block Design (RCBD) with four replications and five feeding supplemented diets with probiotics treatments.

5. Olajide, P.O. Influence of cobalt supplementation on feed intake nutrient digestibility and body weight change in goats fed corncob-based diet. J. Anim. Sci. Livest. Prod. 2017, 1, 003.

Abstract: A 12-week feeding trial was conducted to study the response of West African Male Dwarf goats to a corncob based diet supplemented with varied levels (0.00, 0.50, 0.75 mg/kg DM) of cobalt chloride. Twelve West African Dwarf goats of similar body weights (8.48 ± 1.24 kg) were allotted to three dietary groups, each of four replicates in a randomized complete block design experiment.

  1. Chowdhury, M.R.; Khan, M.M.H.; Mahfuz, S.U.; Baset, M.A. Effects of dietary supplementation of spices on forage degradability, ruminal fermentation, in vivo digestibility, growth performance and nitrogen balance in Black Bengal goat. J. Anim. Physiol. Anim. Nutr. 2018, 120, e591-e598.

Abstract: The objective of the study was to evaluate the effects of spices on forage utilization and nitrogen (N) emission using in vitro and in vivo approach. A 6 × 5 factorial triplicate arrangement was used to assess the in vitro degradability of rice straw with control (without spices) and individual (40 mg/g rice straw) spices (cumin, coriander, clove, black cumin, turmeric) at five different incubation times. In vitro dry matter (DM) and organic matter (OM) degradability of rice straw were highest in presence of spices except for clove. Clove significantly reduced the total volatile fatty acids concentration, molar proportion of acetate and propionate ratio, but increased propionate production. Acetate and butyrate production were not affected by treatments. The ammonia-nitrogen concentration was lowest for clove and turmeric compared to other spices. Rumen pH was unchanged but gradually decreased over the incubation period. For in vivo study, 12 bucks with average live weight 7.65 ± 0.19 kg were assigned to a completely randomized design with three treatments and four replicates for a 28-day period.

  1. Pandey, L.N.; Tiwari, M.R.; Bahadur KC, B.; Baskota, N.; Banjade, J.N. Feeding response of tree fodder Bhimal (Grewia optiva) on growth performance of castrated male goats. J. Nepal Agric. Res. Counc. 2017, 3, 2392-4543.

Abstract: Bhimal (Grewia optiva) is a fodder tree mostly found in mid hills of mid and far western region of Nepal. Bhimal could constitutes one of the main livestock green fodders, especially for goats when fresh green fodder become limited during the winter dry season. However, the feeding value of Bhimal leaves on growth performance of castrated goats probably has not been evaluated so far. Therefore, an experiment was conducted to evaluate the effect of Bhimal leaves feeding on growth performance of castrated male goats for 90 days. Altogether 16 growing castrated male goats of same breed, age and body weight were selected and equally divided into four treatments T1, T2, T3 and T4 with four replications by using Completely Randomized Design (CRD).

  1. Yahya, M.M.; Umar, F.; Yakubu, A.K. Effect of feeding graded levels of probiotic supplemented sugarcane bagasse on performance and haematological parameters of red soko to goats. J. Food Agric. Environ. 2020, 16, 41-48.

Abstract: The aim of the study was to evaluate the effects of probiotic supplemented sugarcane bagasse on performance, haematology, digestibility, and economic analysis of Goats. Sixteen (16) Red Sokoto Goats were assigned to four treatments consisting of four replications in a Completely Randomized Design (CRD). The probiotic supplemented sugarcane bagasse was fed at a concentration of 0, 10, 20, and 30% representing treatments T1, T2, T3, and T4 respectively.

9. Sath, K.; Borin, K.; Preston, T.R. Effect of levels of sun-dried cassava foliage on growth performance of cattle fed rice straw. Livest. Res. Rural. Dev. 2008, 20, supplement. Retrieved January 18, 2023, from http://www.lrrd.org/lrrd20/supplement/sath2.htm

Abstract: An on-farm trial experiment was carried out in Treang district, Takeo province from June to September 2006. Twenty female cattle were allocated to five levels of sun-dried cassava foliage (0, 0.25, 0.5, 0.75 and 1 % of body weight in DM basis) to evaluate the growth response when fed a basal diet of untreated rice straw plus a rumen supplement. The heifers were tethered alongside the feed trough in each household, where they had free access to the experimental diet and water. The heifers were provided rumen supplement (mainly urea, sulphur and other minerals) at 0.25% body weight and ad libitum rice straw. The design was a completely randomized design (CRD) with four replications of each treatment.

  1. Rahman, M.M.; Sarker, N.R.; Alam, M.A. Effect of feeding high yielding fodders on growth performance of growing Hilly Brown Bengal goat. Bang. J. Livs. Res. 2020, 27, 73-81.

Abstract: This study was carried out to evaluate the feeding effect of high yielding fodders (HYF) on feed intake and growth performance of growing Hilly Brown Bengal (HBB) goat. For this purpose, a feeding trail was conducted with 16 growing HBB kids (4 to 5 months) by dividing equally in four groups having four replicates for a period of 75 days. The goats in group T0 (control) received natural grass along with 101.30g concentrates and ad libitum cowpea hay, whereas in group T1, T2 and T3, only natural grass was replaced by BLRI Napier 3, BLRI Napier 4 and Ruzi fodder, respectively.

  1. Ahmed, S.A.; Amodu, J.T.; Abdu S.B.; IShiaku, Y.M.; Lasisi, .O .T.; Abubakar S.A.; Ibrahim, H. Performance of Yankasa rams fed different ratios of Brachiaria ruziziensis forage and concentrate in a total mixed ration . Niger. J. Anim. Sci. Technol. 2019, 2, 63-71. 

Abstract: A feeding trial which lasted for (12) weeks was carried out to determine feed intake, nutrient digestibility and nitrogen balance by growing Yankasa rams fed different ratios of Brachiaria ruziziensis and concentrate in a total mixed ration at the National Animal Production Research Institute Shika, Zaria in Northern Guinea Savanna of Nigeria. Sixteen healthy Growing Yankasa rams with an average live weight of 21.75kg±1kg were randomly allocated to four treatments with four replicates in a Complete Randomized Design (CRD). 

  1. A VFA analysis requires specific information on the device, column, and features used in the chromatographic analysis.
  • Already added information, please see in line 131.

Results:

  1. In the ideal case, Table 5 should display the cost of feed in dollars at the exchange rate at the time the study was conducted.
  • Already changed the information, please see in text.

Discussion:

  1. Yeasts and some bacteria, such as Lentiloactobacillus buchneri, are capable of degrading cell walls. This needs to be referred to in Section 4.1.
  • Already added information, please see in line 192.
  1. It is not stated in the material method that the diets were prepared in an isonitrogenous form. There is also no way to mix 1:1 ratio in order to obtain an isonitrogenous diet compared to the control group (concentrate), as the concentrate feed and fermented cassava peel must have the same chemical composition in order to achieve this.
  • In the present work, we attempt to employ agro-industrial by-products that are abundant in the area by fermenting them with yeast and EM that contain a high concentration of CP in order to reduce feed costs. Although fermented cassava peel does not have the exact same chemical content as concentrate, it is almost equivalent to a diet of concentrate. As the results demonstrated that fermented cassava peel has the potential to be used as a replacement for concentrate, we would confirm the utility of this research.
  1. In section 4.3, it is recommended to omit the statement: " The results indicated that giving goats YFCP or EMFCP instead of concentrate had no effect on the rumen ecology or fermentation in goats". Just by looking at the pH of the rumen, we cannot conclude that it does not affect rumen ecology or fermentation.
  • Already deleted information, please see in the text.

Reviewer 3 Report

Since there are no line numbers, it is difficult to indicate the subject of the comment.

I can understand this experiment, it was possible to conclude that yeast or EM might be utilized as microorganisms to increase the nutritional value of cassava peel. Moreover, YFCP or EMFCP can replace concentrate up to 50 % without impact on feed intake, digestibility, rumen fermentation characteristics and growth performance whereas, can reduced feed cost per gain up to 32 %.

My specific comments are as follows,

Abstract; What are CP, YFCP and EMFCP ?

P2; 65-70% of ruminant production ... cost of ruminant production. --- Words overlap in the same sentence. Please try to work it out.

P2; from 2.4.8% to --- OK?

P2; What are EM, YFCP and EMFCP ?

Table 1. What is EMYFCP ?

Please also list chemical composition for experimental diets, control, YFCP and EMFCP.

P4; SAS software --- What are the version, company name and place ?

P4; The feed --- The feed ingredients

Table 2; in Apparent Digestibility, Dry matter 65.3 63.5 73.1, SEM 2.98 p-value 0.65, Is this OK ?

Organic matter 68.3 66.2 75.6,  SEM 2.97 p-value 0.81, Is this OK ?

Based on the difference in means and SEM values, I would think there would be a significant difference.

Table 2; What are a, b as superscript ?

Table 4; Ruminal volatile fatty acid profile of... --- Ruminal volatile fatty acid (VFA) profile of

in Total VFA, mean, 110.4 104.2 101.5, SEM 2.04, p-value 0.33, Is this OK ?

Based on the difference in means and SEM values, I would think there would be a significant difference.

Table5; in ADG, 0 to 30-d, 60.0 43.3 50.0, SEM 1.65, p-value 0.83, Is this OK ?

Based on the difference in means and SEM values, I would think there would be a significant difference.

Check also in 31 to 60-d.

Is there a note anywhere on how to find the FCR?

P7; The molar proportion of volatile fatty acids, --- The molar proportion of VFA,

P7; What is TMR ?

Author Response

My specific comments are as follows,

  1. Abstract; What are CP, YFCP and EMFCP ?
  • Already added information, please see in the text.
  1. P2; 65-70% of ruminant production ... cost of ruminant production. --- Words overlap in the same sentence. Please try to work it out.
  • Already improved sentence, please see in line 48.
  1. P2; from 2.4.8% to --- OK?
  • Already corrected, please see in line 74.
  1. P2; What are EM, YFCP and EMFCP ?
  • Already added information, please see in line 73, 79.
  1. Table 1. What is EMYFCP ?
  • Already corrected, please see in the text.
  1. Please also list chemical composition for experimental diets, control, YFCP and EMFCP.
  • Table 1 provides the chemical composition of the experimental diets consisting of concentrate (control), YFCP, and EMFCP already; however, for easier understanding, we have shifted the column of experimental diets as shown in the text.
  1. P4; SAS software --- What are the version, company name and place ?
  • Already added information, please see in line 139.
  1. P4; The feed --- The feed ingredients
  • Already improved, please see in line 145.
  1. Table 2; in Apparent Digestibility, Dry matter 65.3 63.5 73.1, SEM 2.98 p-value 0.65, Is this OK ? Organic matter 68.3 66.2 75.6, SEM 2.97 p-value 0.81, Is this OK ? Based on the difference in means and SEM values, I would think there would be a significant difference.
  • We have already double-checked the data and the statistical analysis, and the results are the same as before.
  1. Table 2; What are a, b as superscript ?
  • Already added information, please see in the text.
  1. Table 4; Ruminal volatile fatty acid profile of... --- Ruminal volatile fatty acid (VFA) profile of
  • Already added information, please see in the text.
  1. in Total VFA, mean, 110.4 104.2 101.5, SEM 2.04, p-value 0.33, Is this OK ? Based on the difference in means and SEM values, I would think there would be a significant difference.
  • Already double-checked the data, statistical analysis and improved information, please see in the text.
  1. Table5; in ADG, 0 to 30-d, 60.0 43.3 50.0, SEM 1.65, p-value 0.83, Is this OK ? Based on the difference in means and SEM values, I would think there would be a significant difference. Check also in 31 to 60-d.
  • Already double-checked the data, statistical analysis and improved information, please see in the text.
  1. Is there a note anywhere on how to find the FCR?
  • Already added information, please see in line 116.
  1. P7; The molar proportion of volatile fatty acids, --- The molar proportion of VFA,
  • Already changed, please see in line 224.
  1. P7; What is TMR ?
  • Already added information, please see in line 242.

Round 2

Reviewer 2 Report

Dear Editor and Authors,

Although I argued that not enough animals were used for obtaining a healthy result, the authors were able to persuade me with their explanation. In addition, the article has been revised to address the deficiencies noted in the original version. Greetings to all of the authors who contributed to this work and best wishes for their future endeavors.

Best regards,

Author Response

Thank you very much for your kind comment and suggestion to make the manuscript ready for possible publication in Animals.

Reviewer 3 Report

>Please also list chemical composition for experimental diets, control, YFCP and EMFCP.

>Table 1 provides the chemical composition of the experimental diets consisting of concentrate (control), YFCP, and EMFCP already; however, for easier understanding, we have shifted the column of experimental diets as shown in the text.

What I am trying to say is that I would like you to note not only the ingredients of the individual feeds, but also the chemical composition of the experimental feeds after mixing, control, YFCP, EMFCP, which the animals are actually eating. Otherwise, it will be difficult to discuss.

Author Response

Already added information, please see in Table 2 and line 147 and 206.
